# Low to moderate wave exposure did not impact blue mussel (*Mytilus edulis*) growth in a mesocosm study

**Ivana Lukić** [1,2], **Lucas Hayes** [1,3], **Trine Bekkby** [1] *

**1** Norwegian Institute for Water Research (NIVA), Oslo, Norway, **2** GEOMAR, Helmholtz-Centre for Ocean Research, Kiel, Germany, **3** University of Barcelona, Barcelona, Spain

* trine.bekkby@niva.no

## Abstract

Little is known about the causes of the decline in blue mussel populations in the North Atlantic. If mussel beds are to be protected, and maybe even restored, we need knowledge about environmental conditions under which blue mussels can survive and grow. Wave exposure impacts the growth and abundance of blue mussels by impacting food availability, predation, competition and sedimentation. In the field it is difficult to separate the effects of the different variables, and we therefore wanted to perform a simple, but controlled, mesocosm study on the impact of wave exposure on blue mussel (*Mytilus edulis*) growth. We placed three replicate blue mussels in each of 12 mesocosm basins, of which six had high and six had low wave level. Each of the 36 blue mussels were measured weekly for 13 summer weeks and the measured parameters (length, width, thickness, weight and displacement volume) were analysed against wave exposure and time using a non-parametric Generalised Additive Model (GAM). Surprisingly, we found no effect of wave exposure on any of the parameters. This could be because wave exposure is not as important as we have believed, but that it usually captures other factors, such as sedimentation, predation and competition. It could also be explained by the level and span in wave exposure being too low, failing to generate measurable effects. Our advice for future studies is to increase the difference in wave exposure levels, but still perform controlled studies to separate the effect of wave exposure from other variables.

## Introduction

Blue mussels (*Mytilus edulis*) live from the intertidal down to approximately 30 m depth [1, 2] and are suspension feeders and filter water for microalgae [3]. Blue mussels are exposed to gradients of many interacting environmental conditions, which influence the mussels' survival and growth. These include both abiotic factors, such as salinity, wave exposure, temperature, desiccation and depth, and biotic factors, such as predation and competition for space and food [2, 4].

**Data Availability Statement:** The data underlying the results presented in the study are available from Zenodo, https://doi.org/10.5281/zenodo.13643746.

**Funding:** The project (COASTFRAG) was funded (2021-ongoing) by the Research Council of Norway, grant number 342628/L10 (project number 314314), and the EU grant AQUACOSM-plus (TA-project MESOFRAG, H2020-INFRAIA-Project No 871081). Additional funding was provided by the Norwegian Institute for Water Research. The funders had no role in study design, data collection and analysis, decision to publish, or preparation of the manuscript.

**Competing interests:** The authors have declared that no competing interests exist. All authors declare that the research was conducted in the absence of any commercial or financial relationships that could be construed as a potential conflict of interest.

Wave exposure is a highly important environmental factor on rocky shores, affecting the distribution and abundance of individual species and their interactions [5]. Wave exposure is shown to impact growth and abundance of blue mussels by impacting the food availability [2]. Wave exposure also impacts predation pressure (which is lower at high wave levels, [6]) and competition, e.g., between blue mussel and algae, barnacles or other mussel species [7]. At the same time, high wave exposure might remove sediments [2] and impose a large physical stress to blue mussels, as they need to spend more energy on attaching to the substrate [8]. It is therefore difficult to disentangle the impacts of the different mechanisms through which wave exposure impacts blue mussel growth, abundance and survival.

During the last years, several concerns have been raised regarding declining blue mussel populations in coastal waters, both along the Norwegian coast [9] and other regions in the North Atlantic [10]. In Norway, mussel beds are of conservation concern (vulnerable in the red list of habitats). Little is known about the causes of the decline, though many have been suggested, including climate change, habitat destruction, increased predation and disease [11–13]. If blue mussel beds are to be protected, and maybe even restored, knowledge is needed about the environmental conditions under which the blue mussels have optimal survival and growth. As it is difficult to separate the effects of wave exposure from effects of predation, competition, sedimentation and other variables in the field, we seeked to perform a simple and controlled mesocosm study on the impact of wave exposure on growth of blue mussels, with water quality and wave exposure levels matching those found in the Oslofjord.

## Materials and methods

### The mesocosm basins and experimental setup

This experiment was conducted in 12 hard bottom mesocosm basins at the NIVA Marine Research Station Solbergstrand (59.62˚ N, 10.65˚ E) in the outer Oslofjord, southeast of Norway. Each basin has a volume of 12 m$^3$ and a depth (at high tide) of 1.3 m [14]. The basins are individually supplied with water from the Oslofjord, from 1 m depth just outside of the station. The water has a residence time in the basins of 2–3 hours, which means that the water conditions (temperature and salinity) are similar to the Oslofjord [15]. Tides are simulated with a tidal amplitude of 36 cm. A wave machine produces waves (18 strokes per minute) at two different levels, and half of the basins had high wave level, half of the basins had low wave levels. The high wave level corresponds roughly to what we get from a wind strength of 5 m/s ("gentle breeze" at the Beaufort scale). The low wave level corresponds to a wind strength og 2.5 m/s ("light breeze") [15, 16]. These are common wind levels in this part of the Oslofjord during summer, see e.g., the Gullholmen weather station (SN17280, data on wind rose with frequency distribution, provided by the Norwegian Centre for Climate Services, https://seklima.met.no/).

The blue mussels entered the basins in their pelagic stage in the spring of 2022, as part of the water masses supplied from the fjord outside. When this study started, they were on average 31 mm long (ranging from 27–40 mm). Within each basin, three replicate blue mussels collected from the basins, identity marked, put in a cage (to avoid predation from fish) and hung out just below the low tide level in the basins, ensuring that the mussels were always kept submerged (Fig 1). The blue mussel cages were placed close to the wave machine, ensuring maximum water flow.

All applicable guidelines for sampling, care and experimental use of organisms used in this study have been followed and the necessary approvals have been obtained (from the Norwegian Food Safety Authority, NFSA). No approval from the research ethics committees was required to accomplish the goals of this study because experimental work was conducted with unregulated invertebrate species.

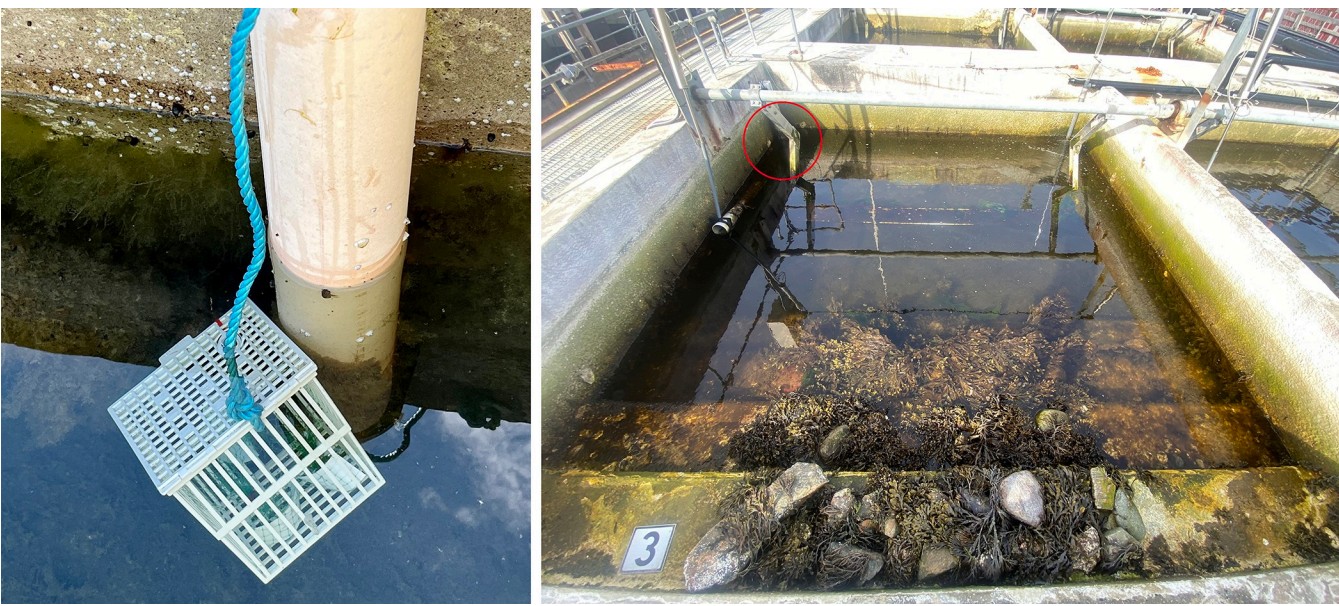

**Fig 1. Example of one of the 12 mesocosm basins.** Three replicate blue mussels (*Mytilus edulis*) were put in a cage (left) and hung out just below the low tide level in the basins (right, red circle).

## Mussel measurements

Each week, over a period of 13 weeks (22.06–20.09.2022), the parameters length, width, thickness, weight and displacement volume of each mussel were measured, after being drip-dried for 2–3 seconds on a paper towel and placed on a scale, where weight was recorded to the nearest centigram. Next, maximum length, width, and thickness were measured to the nearest millimetre using a calliper. Finally, displacement volume was measured to the nearest millilitre in a volumetric flask. Each mussel was then quickly placed back into its net and returned to the basins.

## Statistical analyses

We analysed the blue mussel parameters (length, width, thickness, weight and displacement volume) against wave exposure level and time through a non-parametric Generalised Additive Model (GAM) with wave level as a categorical variable and date as a continuous variable. Basin number was included as a random factor to account for dependencies between the three replicate mussels in each basin and mussel ID was included to identify the parameters belonging to each individual blue mussel. We also analysed the change from the first and last measure, both as change in individual values and as percent changes. The statistical analyses were run using the packages *lme4* [17], *car* [18], and *emmeans* [19] in R [20].

## Results

The parameters length, width, thickness, weight and displacement volume all increased significantly with time (Table 1). These measured parameters correlated to a high degree (Table 2). Based on the correlation values, we chose to present the plot for mussel weight over time as an illustration on growth, as this parameter had the highest correlation with the other parameters. Fig 2 shows the increase in weight over time, for both high and low wave exposure. The weight changed from an overage of 3.2 g (ranging from 2.3 g to 4.4 g) in June to an average of 5.1 g

**Table 1. The statistics from the GAM analyses.**

| Measured parameter | Wave level | Date | Wave:Date interaction | Basin number |
|---|---|---|---|---|
| | t-/p-values | t-/p-values | t-/p-values | F-/p-values |
| Values measured weekly over 13 weeks | | | | |
| Length | 0.466/0.641 | 15.589/**<0.0001** | -0.468/0.640 | 47.47/**<0.0001** |
| Width | 0.211/0.833 | 12.222/**<0.0001** | -0.198/0.843 | 49.23/0.0977 |
| Thickness | 1.515/0.13 | 11.822/**<0.0001** | -1.516/0.13 | 5295/**<0.0001** |
| Weight | 0.419/0.675 | 19.212/**<0.0001** | -0.415/0.678 | 6372/**<0.0001** |
| Displacement volume | 0.948/0.344 | 11.812/**<0.0001** | -0.947/0.344 | 883/**0.008** |
| Percentage (%) change in values from one week to the next over the over 13 weeks | | | | |
| Length change | -0.091/0.927 | -0.095/0.927 | 0.090/0.928 | 0.001/0.375 |
| Width change | 0.102/0.919 | 0.963/0.336 | -0.103/0.918 | 0/0.665 |
| Thickness change | -0.662/0.508 | -0.500/0.618 | 0.662/0.509 | 0/0.742 |
| Weight change | 1.061/0.289 | 2.201/**0.0283** | -1.061/0.2895 | 0.001/0.453 |
| Displacement volume change | 0.725/0.469 | 1.195/0.233 | -0.725/0.469 | 0.006/0.345 |

The analyses were performed on the effect of wave exposure and date on the different parameters measured for the 36 blue mussels (three in each of the 12 basins), weekly over 13 weeks.

(ranging from 3.7 g to 7.4 g) in September. Data for all the measured parameters are presented in the S1 Table and are also available from Zenodo [21]. The percentage increase in most of the parameters did not change over the course of this study (Table 1). An exception was the percentage increase in weight, which significantly changed over time (Table 1). We found no significant effects of wave exposure for any of the measured values, but there was variation between the different basins, regardless of the wave exposure, though not in the percentage change from one time to another (Table 1).

## Discussion and conclusion

During the 13 weeks duration of this study, we found no effect of wave exposure on the development in the different size parameters measured for the blue mussels. We want to point out that we did not aim to obtain growth curves, but rather see if the parameters were impacted by wave exposure at the wave levels found in the Oslofjord. To our surprise, it was not, despite wave exposure being considered a highly important environmental factor on rocky shores [5]. One theory for the negative results found is that there really is no impact of wave exposure on blue mussel growth when other relevant variables, such as predation, competition and sedimentation, are controlled for. If wave exposed areas are also areas with reduced sedimentation and fewer predators and competitors (due to the relatively tough conditions), it is difficult to disentangle the impacts of the different mechanisms through which wave exposure impacts blue mussel growth, abundance and survival. If we look in more detail at the results relating

**Table 2. The correlation matrix of the different parameters measured.**

| | Length | Width | Thickness | Weight | Displacement volume |
|---|---|---|---|---|---|
| Length | 1.00 | 0.81 | 0.66 | 0.90 | 0.80 |
| Width | | 1.0 | 0.59 | 0.85 | 0.71 |
| Thickness | | | 1.00 | 0.79 | 0.67 |
| Weight | | | | 1.00 | 0.83 |
| Displacement volume | | | | | 1.00 |

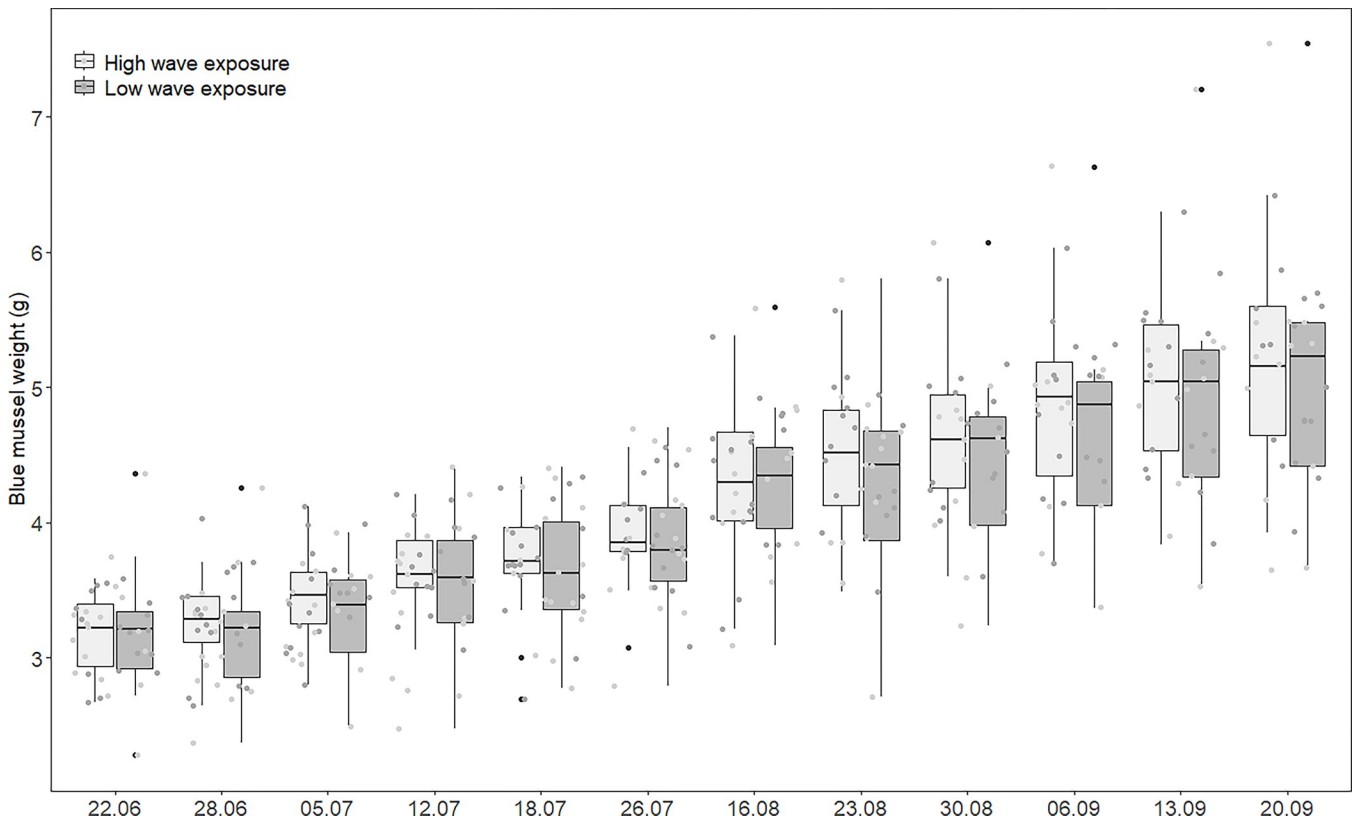

**Fig 2. The weight of the blue mussels with time.** Boxplot of the weight of the blue mussels (*Mytilus edulis*) over a period of 13 weeks (22.06–20.09.2022). The growth was monitored in six basins with high and six basins with low wave exposure, presented here as light grey (high wave) and dark grey (low wave) boxes and points. Each point in the plot represents an individual blue mussel. Horizontal lines: median value, box: the 25–75 quartile, whiskers: data outside of the middle 50%, dots: outliers (i.e., the black dots: 1.5 times the interquartile range). The grey dots show the data points.

waves to mussel growth [2], the wave exposure was also related to the amount of sediment [22]. Wave exposed areas also provide refuges from predation [6, 23] and impacts the competition, e.g., between blue mussel and algae [7], which also makes it difficult to identify the impact of wave exposure alone. In our study, the blue mussels were protected from such co-varying factors, indicating that wave exposure might not impact blue mussel growth when other factors are accounted for. However, our study might have failed to generate measurable effects because of the low level and narrow span in wave exposure in our basins. The "high wave" basins had a wave level corresponding roughly to "gentle breeze" (at the Beaufort scale). "Low wave" corresponds to "light breeze". It is difficult to compare our wave levels with that of other studies, as the methods and units used for assessing wave exposure vary a lot. However, several studies have found faster growth in mussels and other intertidal filter-feeders in wave exposed compared to sheltered areas [24–26], and effects of wave exposure on mussels were reported in an area that seemed to have a wider wave gradient than we had, with no co-varying factors were reported [8].

Our study suggests that the impact of wave exposure on mussel growth that is often found in different studies may be more due to the waves' influence on processes such as sedimentation, predation and competition than due to factors more directly related to wave exposure. However, we do believe that impacts of waves can be found under controlled conditions, as long as the wave levels are sufficient. One factor more directly linked to wave exposure is the access to food [2, 27], i.e., microalgae [3]. However, extreme wave exposure levels reduce the

growth of mussels, most likely due reduced filtration ability the use of energy for attachment (byssus production [8]). In our study, we did not monitor the phytoplankton concentrations, so we have no knowledge of the food availability in the basins. If the food was sufficiently abundant in all basins, the difference between the two wave levels might not be sufficient to impact the food availability. Our advice for further studies is to increase the difference in wave exposure level, while also testing for different levels of food availability. We advise doing this controlling for predation, competition, sedimentation and other factors that might impact blue mussel growth. If we understood more about the environmental conditions under which the blue mussels have optimal survival and growth, we would have a better base line for protecting and restoring blue mussel beds.

## Supporting information

**S1 Table. Data on length, width, thickness, weight and displacement volume of blue mussel (Mytilus edulis) measured weekly in a mesocosm study over 13 weeks in 2022.** BN = basin number; MN = mussel number; WE = wave exposure level; L = length (mm); LC = change in length from one week to the other, for each mussel in each basin; WI = width (mm); WIC = Width change; T = thickness (mm); TC = thickness change; WE = weight (mm); WEC = weight change; DV = displacement volume; DVC = displacement volume change. (DOCX)

## Acknowledgments

Thanks to Benjamin Kupilaas (NIVA) for all the help with the communication towards the AQUACOSM-plus program. The study took place at the NIVA Solbergstrand Experimental Facility, and we want to thank all the staff at Solbergstrand for their help and support. We also want to thank the reviewers for all the useful comments and suggestions for changes.

## Author Contributions

**Conceptualization:** Trine Bekkby.

**Data curation:** Ivana Lukić, Lucas Hayes.

**Formal analysis:** Ivana Lukić.

**Funding acquisition:** Trine Bekkby.

**Investigation:** Ivana Lukić, Lucas Hayes.

**Methodology:** Trine Bekkby.

**Project administration:** Trine Bekkby.

**Supervision:** Trine Bekkby.

**Writing – original draft:** Ivana Lukić, Lucas Hayes, Trine Bekkby.

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
