## [Decision Letter · Decision Letter 0]

27 Aug 2024

PONE-D-24-25103Low to moderate wave exposure did not impact blue mussel (Mytilus edulis) growth in a mesocosm studyPLOS ONE

Dear Dr. Bekkby,

Thank you for submitting your manuscript to PLOS ONE. After careful consideration, we feel that it has merit but does not fully meet PLOS ONE’s publication criteria as it currently stands. Therefore, we invite you to submit a revised version of the manuscript that addresses the points raised during the review process.

**Please review the manuscript. Both reviewers have valuable comments, especially Reviewer #2 regarding the interpretation of the data and the data availability.**

We look forward to receiving your revised manuscript.

Kind regards,

José A. Fernández Robledo, Ph.D.

Academic Editor

PLOS ONE

**Journal Requirements:**

The project (COASTFRAG) was funded (2021-ongoing) by the Research Council of Norway, grant number 342628/L10 (project number 314314), and the EU grant AQUACOSM-plus (TA-project MESOFRAG, H2020-INFRAIA-Project No 871081). Additional funding was provided by the Norwegian Institute for Water Research. 

Thanks to the Research Council of Norway, the EU project AQUACOSM-plus and NIVA for funding. We are grateful to Benjamin Kupilaas (NIVA) for all the help getting the AQUACOSM-plus grant. The study took place at the NIVA Solbergstrand Experimental Facility, and we want to thank all the staff at Solbergstrand for their help and support. 

The project (COASTFRAG) was funded (2021-ongoing) by the Research Council of Norway, grant number 342628/L10 (project number 314314), and the EU grant AQUACOSM-plus (TA-project MESOFRAG, H2020-INFRAIA-Project No 871081). Additional funding was provided by the Norwegian Institute for Water Research. 

Reviewers' comments:

Reviewer's Responses to Questions

**Comments to the Author**

1. Is the manuscript technically sound, and do the data support the conclusions?

Reviewer #1: Yes

Reviewer #2: Partly

2. Has the statistical analysis been performed appropriately and rigorously? 

Reviewer #1: Yes

Reviewer #2: Yes

3. Have the authors made all data underlying the findings in their manuscript fully available?

Reviewer #1: No

Reviewer #2: No

4. Is the manuscript presented in an intelligible fashion and written in standard English?

Reviewer #1: Yes

Reviewer #2: Yes

5. Review Comments to the Author

**Reviewer #1:** Review of Lukic et al et al. “Low to moderate wave exposure did not impact blue mussel (Mytilus edulis) growth in a mesocosm study”, submitted to PLOS One (PONE-D-24-25103).

The paper entitled “Low to moderate wave exposure did not impact blue mussel (Mytilus edulis) growth in a mesocosm study”, relates to a very simple study done in a mesocosm experiment in order to test the effect of different wave exposures on the mussel growth. The topic of study is not novel, but deserves our attention.

General Comments:

The paper is very concise and the results were not the expected according to the authors, however the experiment was run properly, despite the paper can be improved. I recommend major revisions.

Specific Comments:

1) In the materials and methods is not well described the experimental set-up. In the abstract the authors mention that from the 12 mesocosms, 6 were for the “high wave exposure” while the other 6 were for the “low wave exposure”, but in the M&M this is not mentioned. Also, in this section is not described the size or age of the mussels used for the experiment. This should be described also.

2) In the results section, surprisingly, are not shown any results, including graphs with the evolution of mussel measurements for the different treatments. It is just represented the statistical results. But in my opinion, it is missing some graphical representation of the results, that can be complemented with the stats.

3) In the stats tables, I do not understand the advantage of studying the percentage change of the different parameters from one time to the next over the 13 weeks (table 1) and from the start to the end of the study (table 2). This should be better explained.

4) The discussion is quite superficial. The authors could improve it with other case studies to compare with.

5) Figure1 legend appears in the middle of the text, apart from the figure. Please check.

6) In the discussion, line 130, please delete the word “these”. It should be read “influence on other processes” in spite of “influence on these other processes”.

7) Line 135, the sentence is confusing. Please rephrase.

Concluding, I suggest that this manuscript can be published in PLOS One after major revisions.

**Reviewer #2: **The manuscript summarizes the results of a set of experiments on the effects of wave exposure on the growth of the mussel *Mytilus edulis* in mesocosms. The authors found no significant differences between treatments. They attribute the lack of an effect to the low strength of the simulated waves.

The experimental design and statistical methods appear to be sound, but I have two major concerns about the manuscript:

My first concern is the absence of results in the manuscript. I am not referring to the lack of significant differences; publishing negative results is valuable. The problem is that none of the data from the experiments appear in the paper, either in table or plot form. Statistical tests are presented, but these are essentially meaningless without seeing the values on which they are based. The manuscript cites a link to a Zenodo database, but these data do not seem to be available to reviewers, and they should be included within the text of the manuscript rather than via an external link, preferably in the form of plots. The authors write that they "did not aim to study the growth patterns of blue mussels", but this is what was measured, and the results are needed to understand the conclusions of the manuscript and could also be useful in other contexts.

My second concern is with the interpretation of the results. There are two alternative hypotheses for the negative results presented. The first is that there really is no effect of wave exposure on mussel growth when the effects of other relevant variables, including predation and food availability, are removed. The second is that the waves used in the study were not strong enough to generate a measurable effect. These two alternatives should be explicitly explored in the discussion and conclusion section and included in the abstract. There is likely information in the literature supporting one hypothesis over the other. For example: How did the strength of your wave exposures compare to those from previous studies? How did phytoplankton concentrations compare? It is worth exploring this further because if there really is no effect of waves on the growth absent the other factors, this could be an interesting finding. Overall, more effort should be devoted to putting the results in the context of previous work.

I believe that addressing these two concerns could produce a much stronger manuscript. While no significant differences were found between treatments, this can still be an interesting and useful result if presented in the right way.

Minor comments:

Line 21: ..."a" simple, but controlled, mesocosm study...

Line 45: Should this be "remove sediment" instead of "remove sedimentation"?

Line 129: ...impact of "wave" exposure...

6. PLOS authors have the option to publish the peer review history of their article (what does this mean?). If published, this will include your full peer review and any attached files.

Reviewer #1: No

Reviewer #2: No

---

## [Author Response · Author response to Decision Letter 0]

4 Sep 2024

Thanks for the thorough evaluation of our manuscript and all the suggestions for improvement, we highly appreciate it. We have gone through all the comments, questions and suggestions, and answered them, one by one. I hope our response is to your satisfaction. The response to reviewers is uploaded as a separate file.

---

## [Decision Letter · Decision Letter 1]

1 Oct 2024

PONE-D-24-25103R1Low to moderate wave exposure did not impact blue mussel (Mytilus edulis) growth in a mesocosm studyPLOS ONE

Dear Dr. Bekkby,

Thank you for submitting your manuscript to PLOS ONE. After careful consideration, we feel that it has merit but does not fully meet PLOS ONE’s publication criteria as it currently stands. Therefore, we invite you to submit a revised version of the manuscript that addresses the points raised during the review process.

Dear Dr. Bekkby,

I do acknowledge your efforts answering the reviewer's comments. I understand that the authors should consider to include in the main body of the manuscript the data plots as suggested by both reviewers. These plots will add the context for the interpretation and discussion of the results.

We look forward to receiving your revised manuscript.

Kind regards,

José A. Fernández Robledo, Ph.D.

Academic Editor

PLOS ONE

Reviewers' comments:

Reviewer's Responses to Questions

**Comments to the Author**

1. If the authors have adequately addressed your comments raised in a previous round of review and you feel that this manuscript is now acceptable for publication, you may indicate that here to bypass the “Comments to the Author” section, enter your conflict of interest statement in the “Confidential to Editor” section, and submit your "Accept" recommendation.

Reviewer #1: (No Response)

Reviewer #2: (No Response)

2. Is the manuscript technically sound, and do the data support the conclusions?

Reviewer #1: Yes

Reviewer #2: No

3. Has the statistical analysis been performed appropriately and rigorously? 

Reviewer #1: Yes

Reviewer #2: I Don't Know

4. Have the authors made all data underlying the findings in their manuscript fully available?

Reviewer #1: Yes

Reviewer #2: Yes

5. Is the manuscript presented in an intelligible fashion and written in standard English?

Reviewer #1: Yes

Reviewer #2: Yes

6. Review Comments to the Author

Reviewer #1: Review of Lukic et al et al. “Low to moderate wave exposure did not impact blue mussel (Mytilus edulis) growth in a mesocosm study”, submitted to PLOS One (PONE-D-24-25103).

The revised version of the paper entitled “Low to moderate wave exposure did not impact blue mussel (Mytilus edulis) growth in a mesocosm study”, seems to satisfy almost all points mentioned by the reviewers, however, I still recommend in the results section that the authors create a figure, including 4-5 graphs showing the evolution of the mussels measurements through time comparing the high wave treatment with the low wave treatment. Because the table S1 in the supplementary material is quite long, no one will pay attention to that. And in fact, despite there are no differences between the treatments, those are the results obtained and makes sense to present the results and not just the statistics. So, before the paper be accepted, I would like to recommend to introduce that figure in the results section.

Concluding, I suggest that this manuscript can be published in PLOS One after minor revisions.

Reviewer #2: The authors did not address what I believe is the most important issue with this manuscript, the lack of data plots. Providing the raw data is not enough. Reviewer #1 and I both raised this issue in our initial reviews.

7. PLOS authors have the option to publish the peer review history of their article (what does this mean?). If published, this will include your full peer review and any attached files.

Reviewer #1: No

Reviewer #2: No

---

## [Author Response · Author response to Decision Letter 1]

10 Oct 2024

Dear reviewers, 

Thanks for your reply to our manuscript. Both reviewers still recommend that we create a figure, and some text to follow the figure, on the change in the mussel measurements through time comparing the high wave treatment with the low wave treatment. We agree on this, and we have included a figure on changes is weight over time. We started by making a correlation matrix and found weight to be the parameter that correlated the most with the others. We believe that it is enough to include one figure, and weight was therefore selected. If you want us to include a similar figure on all parameters, we will do that. The figure shows the difference between the high and the low wave basins. 

I am sorry for missing to write the repository name and DOI number of the dataset in the Data Availability Statement. The DOI has been included in the reference list in the manuscript and is now also stated in the Data Availability Statement.

Regards, Trine

---

## [Decision Letter · Decision Letter 2]

14 Nov 2024

PONE-D-24-25103R2Low to moderate wave exposure did not impact blue mussel (Mytilus edulis) growth in a mesocosm studyPLOS ONE

Dear Dr. Bekkby,

Thank you for submitting your manuscript to PLOS ONE. After careful consideration, we feel that it has merit but does not fully meet PLOS ONE’s publication criteria as it currently stands. Therefore, we invite you to submit a revised version of the manuscript that addresses the points raised during the review process.

**Dear Dr. Bekkby,****Please edit the figure 2 following reviewer 2 suggestions.** **Sincerely,**

We look forward to receiving your revised manuscript.

Kind regards,

José A. Fernández Robledo, Ph.D.

Academic Editor

PLOS ONE

Journal Requirements:

Reviewers' comments:

Reviewer's Responses to Questions

**Comments to the Author**

1. If the authors have adequately addressed your comments raised in a previous round of review and you feel that this manuscript is now acceptable for publication, you may indicate that here to bypass the “Comments to the Author” section, enter your conflict of interest statement in the “Confidential to Editor” section, and submit your "Accept" recommendation.

Reviewer #1: All comments have been addressed

Reviewer #2: (No Response)

2. Is the manuscript technically sound, and do the data support the conclusions?

Reviewer #1: Yes

Reviewer #2: Yes

3. Has the statistical analysis been performed appropriately and rigorously? 

Reviewer #1: Yes

Reviewer #2: Yes

4. Have the authors made all data underlying the findings in their manuscript fully available?

Reviewer #1: Yes

Reviewer #2: Yes

5. Is the manuscript presented in an intelligible fashion and written in standard English?

Reviewer #1: Yes

Reviewer #2: Yes

6. Review Comments to the Author

**Reviewer #1: **The final revised version of the paper entitled "Low to moderate wave exposure did not impact blue mussel (Mytilus edulis) growth in a mesocosm study" can be accepted for publication in PLOSONE.

Best Regards

**Reviewer #2: **Please use different symbols/colors to distinguish the data points from the two treatments in figure 2. The caption for figure 2 should identify what each data point represents (I am assuming each one is an individual mussel).

7. PLOS authors have the option to publish the peer review history of their article (what does this mean?). If published, this will include your full peer review and any attached files.

Reviewer #1: No

Reviewer #2: No

---

## [Author Response · Author response to Decision Letter 2]

18 Nov 2024

Dear reviewers, 

We have now made the changes that reviewer 2 suggested by changing the colours of the datapoints so that it is easier to see which data point belongs to the two different treatments, high and low wave level. The figure text has been changed to explain this. We have also made it clear in the figure text that each data point represents an individual mussel. 

Regards, Trine Bekkby

---

## [Editor Report · Decision Letter 3]

21 Nov 2024

Low to moderate wave exposure did not impact blue mussel (Mytilus edulis) growth in a mesocosm study

PONE-D-24-25103R3

Dear Dr. Bekkby,

We’re pleased to inform you that your manuscript has been judged scientifically suitable for publication and will be formally accepted for publication once it meets all outstanding technical requirements.

Kind regards,

José A. Fernández Robledo, Ph.D.

Academic Editor

PLOS ONE

Additional Editor Comments (optional):

Thank you for addressing the last point regarding Figure 2 and Figure 2 legend.

Sincerely,

-j

---

## [Editor Report · Acceptance letter]

26 Nov 2024

PONE-D-24-25103R3 

PLOS ONE

Dear Dr. Bekkby, 

I'm pleased to inform you that your manuscript has been deemed suitable for publication in PLOS ONE. Congratulations! Your manuscript is now being handed over to our production team.

Kind regards, 

on behalf of

Dr. José A. Fernández Robledo 

Academic Editor

PLOS ONE